# Delivering culturally adapted family interventions for people with schizophrenia in Indonesia: A feasibility randomised controlled trial and nested process evaluation

**Laoise Renwick**[1]*, **Helen Brooks**[1], **Budi-anna Keliat**[2], **Dewi Wulandari**[2], **Rizqy Fadilah**[2], **Raphita Diorarta**[2], **Suherman**[2], **Georgia Addison**[3], **Penny Bee**[1], **Karina Lovell**[1], **Herni Susanti**[2]

1 Division of Nursing, Midwifery and Social Work, Faculty of Medicine, Biology and Health, University of Manchester, Manchester, United Kingdom, 2 Faculty of Nursing, Universitas Indonesia, Jarkarta, Indonesia, 3 Mental Health Nursing Research Unit, Greater Manchester Mental Health Trust, Manchester, United Kingdom

* Laoise.renwick@manchester.ac.uk

## Abstract

Schizophrenia is a severe and enduring illness with high relapse rates leading to functional impairment. Although family interventions effectively reduce relapse, most evidence originates from high-income settings. This single-masked feasibility trial randomised 74 service-user–carer dyads to receive either a culturally adapted family intervention or treatment as usual. The intervention was delivered by non-specialist healthcare workers through a task-shifting approach integrated into primary care. Feasibility outcomes indicated high recruitment and retention rates, strong intervention fidelity, and good acceptability among participants and facilitators. The process evaluation identified practical enablers and barriers to delivery and confirmed the feasibility of training and supervising non-specialist workers to implement the intervention. Findings provide confidence in progressing to a definitive trial based on the feasibility of recruiting participants and therapists and obtaining outcome measures at end point. Findings indicated there is potential for scaling culturally adapted family interventions within primary healthcare systems in low-resource settings. Further research should focus on refining measurements to ensure consistency and validity in trial methods, and explore factors related to therapist and implementation contexts to understand core and peripheral elements of culturally adapted interventions to optimise effectiveness and understand factors linked with implementation, policy and affordability benefits of improving access to mental healthcare.

**Data availability statement:** All relevant data are within the paper and its Supporting Information files that are available from the Figshare repository at https://doi.org/10.48420/30540854.

**Funding:** This paper presents independent research funded by the Medical Research Council (MR/T003987/1) under its Joint Global Health Trials Funding Scheme titled 'Reducing Relapse for People with Schizophrenia in Jakarta, Indonesia: Developing a culturally-relevant, evidence-based Family Intervention'. This represents joint funding from the Department of Health and Social Care (DHSC), the Foreign, Commonwealth & Development Office (FCDO), the Medical Research Council (MRC) and Wellcome Trust. The funders had no role in study design, data collection and analysis, decision to publish, or preparation of the manuscript.

**Competing interests:** The authors have declared that no competing interests exist.

## Introduction

Schizophrenia, and psychoses, are linked with substantial morbidity and mortality. People with schizophrenia are more likely to experience multiple long-term conditions, die 20 years earlier than the general population, on average, and can experience considerable detriment to their functioning and quality of life due to continuing symptoms [1–3]. Schizophrenia is a severe and enduring illness, and despite heterogenous outcomes, the condition is characterised by high rates of relapse particularly in the early course of illness [4] which further impairs functioning and compromises recovery [5–7]. Consequently, these conditions can have a markedly negative impact on individual social connections, occupational and role functioning and compromise individual ability to maintain independence [3,8]. Additionally, these illnesses confer a profound economic burden to health care globally through increased direct healthcare costs and increased need for care and loss of opportunity [9,10].

In low- and middle-income countries (LMICs), the disease burden is magnified due to sizeable treatment gaps and a near-absence of access to quality care [11]. Schizophrenia is a priority condition, identified by the WHO Mental Health Gap Programme (mhGAP) [12] and the World Bank [13]. Treating schizophrenia via enhancing capacity for improved treatments and increasing access to care is strategically important to meet international policy recommendations. Family interventions (FI) are one of three interventions recommended for treating schizophrenia in the World Bank Disease Control Priorities, based on the strength of evidence supporting their ability to reduce illness burden [14]. FI have a particularly strong evidence base for reducing relapse risk with medium to large effect sizes [14,15] and WHO international clinical guidelines also recommend FI as an essential intervention in LMICs [16]. Despite this, there is minimal evidence of their effectiveness in low resource settings and while it is encouraging that some evidence is emerging, the evidence that is available is both low quality and affected by bias [17].

FI require adaptation to different cultural contexts allowing cultural beliefs, explanatory models of illness and contextual socio-economic issues to be incorporated into the content and delivery of such interventions [18,19]. Evidence from several meta-analytic reviews show culturally adapted psychosocial interventions, such as FI, have demonstrable effectiveness, over and above the benefits achieved by standard, non-tailored treatments [18–20]. Indeed, when effective interventions are successfully adapted, people are more likely to engage with psychiatric help offered and the acceptability of the intervention is increased [19,21]. Our knowledge of the effect of adapted interventions comes largely from studies that have been conducted on minority populations within High-Income-Countries (HIC). Rather, there is a need to evidence the effect of adapted interventions in low-resource settings particularly as evidence from adapted FI delivered to minority groups cannot be reliably generalised to LMICs due to variation in socio-cultural milieus, family structure, social support [22], variable disease patterns and illness trajectories [11]. Equally, evidence from FI delivered to wider populations in high income settings suffers similar generalisability issues and there is a need to provide evidence that is relevant to the context in which interventions will be delivered. Moreover, randomised trials, which are fewer in

low-resource settings, are essential to ensure research is relevant in countries with different disease burdens to effectively tackle global health disparities.

Our research is focused on Indonesia. This archipelagic region has the sixth largest number of people living in extreme poverty compared to other countries globally and income inequality has recently grown faster than in any other country in South-East Asia [23]. In 2020, the WHO South-East Asia Region had the lowest estimated median government expenditure per capita on mental health, which was 75 times less than the global average [24]. National-level health spending in Indonesia is steadily increasing but based on WHO estimates, mental health expenditure at 2% of the total health expenditure, is a negligible percentage of GDP [24]. Mental health is becoming a priority within national health policy but the ratio of community centres and trained healthcare workers to population served remains low by standards set by the WHO and lags other countries in the South-east Asian region [25]. There are estimates that up to 50% of people with schizophrenia receive basic care in primary health centres but also widespread variation in treatment rates and persistent human rights abuses [25].

To our knowledge, there are no comprehensive, adapted FIs for people with schizophrenia and their caregivers in Indonesia. A single study evaluating brief caregiver psychoeducation delivered to families of individuals with early psychosis demonstrated some significant effects, though the outcomes were limited to knowledge and symptoms in a relatively homogenous group at the earliest stage of diagnosis [26]. Evidence from recent scoping [27] and meta-analytic review of FI specifically [15] confirm there is a dearth of evidence to support the delivery and scalability of these interventions for people with schizophrenia in Indonesia. We aimed to investigate the feasibility of delivering culturally adapted FI to families of people with schizophrenia in Indonesia to understand the tasks, resources and processes involved in providing FI in a large-scale trial enabling identification of mechanisms of change through which culturally relevant FI may lead to real-world change. We used systematic, theoretically driven empirical research to design a culturally relevant intervention for families of people with schizophrenia and both the protocol and development work have been published elsewhere [28,29]. Our purpose was to investigate potential obstacles in order that these could be factored into research planning for future effectiveness studies while engaging wider systems to ensure our research is context-relevant [30].

### Aim and objectives

The development of our culturally adapted intervention and the protocol for our research are published elsewhere [28,29,31]. Briefly, our evidence-based FI comprised psychoeducation and cognitive behavioural approaches, which were culturally adapted using multiple stakeholder views. Our intervention was delivered by non-specialist healthcare professionals via task-shifting in primary care settings. The feasibility of delivering our intervention in a future trial was determined by evaluating the (i) acceptability and satisfaction of participants and family workers, (ii) recruitment, attendance and retention in the intervention, (iii) completion of outcome measures pre- and post-intervention, (iv) recruitment of healthcare workers as therapists and (v) fidelity to the intervention. We conducted a nested qualitative process evaluation, guided by the Medical Research Council guidance on conducting process evaluations [32], to explore (i) carers views of receiving the intervention including the content and delivery of each session and acceptability, (ii) carers perceptions of the usefulness of the intervention, cultural relevance and appropriateness of session content and accessibility of the resources used, (iii) factors that influence the delivery of the intervention in practice from the perspective of carers and healthcare professionals (iv) views about the usefulness, relevance, and appropriateness of training for trial therapists in delivering the intervention and (v) carers views of participating in research including participating in measurement and being randomised.

### Methods

### Study design

We conducted a single-masked feasibility trial, randomising service-user-carer dyads in a two-arm, parallel randomised controlled trial to receive the family intervention or treatment as usual in an allocation ratio of 1:1. We conducted a process

evaluation [32] using both qualitative and quantitative measures alongside the feasibility RCT to examine how the intervention was implemented, assess delivery mechanisms and fidelity and evaluate barriers and facilitators to successful implementation to contribute to knowledge of programme success and practicality as well as support the application of the intervention in the future. Randomisation was conducted using sequentially numbered, opaque, sealed envelopes with single random numbers generated manually. Envelopes were prepared by a member of the study team (H.S.) independent of the team member responsible for recruitment (B.A.K.) and contained a pre-determined group assignment prepared in accordance with the randomisation schedule which was designed once all participants were recruited. The schedule allocated number sequences between 01 and 74 at random and envelopes containing assignments were revealed once the participant was ready to be allocated to a treatment arm. Allocation was concealed from assessors but due to the nature of the intervention, neither participants nor healthcare workers could be masked to arm allocation. We asked participants not to disclose their allocation to assessors during outcome evaluations. The trial was registered prospectively (https://doi.org/10.1186/ISRCTN49498363) and is reported here using updated CONSORT guidelines [33].

Briefly, our intervention comprised i) a training manual comprising culturally adapted intervention components (gathering information, providing feedback, communication skills, stress management, problem-solving & coping skills and relapse prevention) separated into workshops and accompanying evidence base and culturally acceptable teaching and learning activities, ii) psychoeducational booklets detailing myths and misconceptions linked with specific family intervention components and iii) a supervision framework to support therapists to deliver family interventions consistently and confidently. The development of our intervention is described elsewhere [31]. Treatment as usual mainly comprises minimal contact with primary care services for consultation and repeat prescriptions.

## Eligibility and inclusion criteria

Participants comprised service users, their carer's and relatives and healthcare professionals recruited from two district primary care centres in Jakarta and Bogor and one tertiary hospital in Bogor, Indonesia. Jakarta is the national capital and administrative centre of Indonesia, characterised by rapid economic growth and significant social inequalities. Bogor lies within the Jakarta metropolitan area but is a distinct city characterised by rapid urbanisation. Both cities are densely populated but there are different health systems, care quality and access. Service-user participants were eligible if they met pre-defined inclusion criteria including having a diagnosis of schizophrenia or related psychosis, currently receiving treatment in a primary care setting, being over the age of 18 and being able to give informed consent as judged by the referring healthcare worker. Eligible carers were those who agreed to participate and if they were living with or spending at least 10 h per week in face-to-face contact with their relative with schizophrenia, were aged 18 and were able to give informed written consent. Healthcare workers were included if they had a permanent contract in a primary care centre in Jakarta or Bogor, had primary responsibility for delivering the mental health programme and could commit to delivering family interventions to a minimum of one service-user/carer dyad.

## Success criteria

We intended to assess the feasibility, relevance, safety, and acceptability of our culturally adapted intervention and research procedures in preparation for a fully-powered randomised controlled trial. As recommended in the CONSORT extension for feasibility studies [34], the number of participant dyads recruited to our study was based on feasibility estimates and informed our decision process about whether to proceed to a full trial. When designing the feasibility randomised trial with feasibility parameters related to recruitment and completion of measures, these included:

1. Recruitment target: 2 dyads/families per centre per month (3 centres)

2. Minimum of 1 healthcare worker per site to deliver the intervention

3. Completion of ~80% of outcome measures (baseline, post-intervention, 3 months) [28]

These criteria were reviewed and approved by the Trial Steering Committee. The sample size was determined to ensure that key feasibility metrics could be estimated with sufficient precision over the 6-month recruitment period. Allowing for potential drop-out and incomplete data, recruiting 36 dyads per arm, with an anticipated 80% completion rate, was expected to yield approximately 29 dyads with complete data. For feasibility assessment, targeting 30 participant dyads per arm (total = 60) was considered appropriate. Defining feasibility success as ≥80% completion would provide a 61% probability of proceeding to a full trial, whereas a lower completion rate of 60% would correspond to a < 5% probability of proceeding. Thus, a sample of 30 dyads per arm was estimated to balance precision, minimise the false-go rate [35], and align with the projected recruitment capacity of approximately two dyads per site per month.

## Participants

All participants received information about the trial, including data collection procedures, details of involvement including the possibility of random selection to either continued treatment as usual or family interventions. The study was granted ethical approval from the University of Manchester Ethical Review Committee (Ref: 2020-8041-13687) and Universitas Indonesia Ethics Committee (Ref: 162/UN2.F12.D1.2.1/PPM2021). Consent was obtained for each participant via written or e-consent forms using Zoho. Consent procedures aligned with local ethical and information governance regulations. Trial data were collected between 1st October 2022 and 30th September 2023.

## Outcomes and measures

Clinical and process outcome data were collected from both family members and patients with schizophrenia at baseline, post-intervention and three months post-intervention. The feasibility study determined whether it would be possible to deliver family interventions via task-shifting to non-specialist healthcare workers in primary care settings. Feasibility measures comprised evaluations of acceptability, recruitment, retention and fidelity to intervention components and delivery. Secondary outcomes relating to the feasibility of obtaining relevant measures for a full-scale trial comprise assessments of symptom severity and relapse rates, social functioning, family functioning and environment and therapeutic engagement.

### Psychosis symptom severity and relapse (service-user)

Clinical symptoms were evaluated using the PANSS [36]. The validity and reliability of the translated Indonesian version have been established [37]. Relapse is operationalised as an increase from mild or below to severe or very severe on one of the following symptoms rated using the PANSS: unusual thought content, hallucinations and conceptual disorganisation for a minimum duration of 1 week [37]. The primary outcome is psychosis relapse. We conducted separate analyses on our primary outcome measure; relapse to ascertain the rate of relapse at each measurement point using two separate relapse definitions based on increase in PANSS total score >=12 points and>=1 point increase together with a score of >4 in>=1 item of the positive and disorganised scales used in the RSWG for schizophrenia definition of remission [38]. The PANSS was administered by specially trained research assistants at baseline, immediately after the intervention and 3 months after the intervention. We ascertained inter-rater reliability between raters on videotaped interviews with psychiatric registrars and actors during training.

### Caregiver psychological wellbeing (caregiver)

The 12-item General Health Questionnaire (GHQ-12) [39] was developed to screen for non-specific psychiatric morbidity, to determine whether an individual is at risk of developing a psychiatric disorder and has been widely validated and found to be reliable. The Indonesian-language version of the GHQ-12 has been tested for reliability and validity, demonstrating good consistency and sensitivity [40].

### Caregiver burden (caregiver)

The Involvement Evaluation Questionnaire (IEQ) [41] is a 31-item questionnaire comprising items relating to tension, supervision, worrying and urging and the degree to which the caregiver has experienced any of these. The scale has been developed for European settings, though it has been translated and validated in several European countries and in LMIC settings [42]. Carer burden was also measured using the Experience of Caregiving Inventory which captures a wider set of negative subscales comprising difficult behaviours, negative symptoms, stigma, problems with services, effects on the family, loss and need for backup. There are two positive subscales consisting of positive personal outcomes and good aspects of the relationship with the patient about the carer's experiences. This measure has been used with a variety of carers of mental health conditions, and each subscale has been reported to have satisfactory reliability [43].

### Family functioning and environment (caregiver)

Expressed emotion (EE) was measured using the Family Questionnaire (FQ) [44]. The questionnaire comprises 20 items each measured on a 4-point scale and consists of two subscales assessing both emotional over-involvement and critical comments. The FQ has excellent psychometric properties including a clear factor structure, good internal consistency of subscales and good inter-rater reliability in relation to the Camberwell Family Interview (CFI) [45] which is the gold standard measure of EE and is sensitive to predicting components of EE.

### Knowledge (caregiver)

Knowledge about schizophrenia and psychosis was measured among participants using the Knowledge About Schizophrenia Test (KAST) [46] which was developed for caregivers of people admitted to hospital for treatment of psychosis. The test comprises 21 items regarding the aetiology, onset, symptomatology, outcome and treatment options. The measure shows excellent content validity and good criterion validity.

### Cultural relevance of measures

The KAST, GHQ, FQ, ECI and IEQ, were cross-culturally adapted following similar procedures outlined by Knudsen et al. [47], comprising translation, back-translation and checking through inquiry with research assistants and participants in the feasibility study about their perception of scale and item comprehension and content validity. Translation was conducted by researchers within the study team, and back-translation was conducted by an independent translation service.

### Training

A 4.5-day training course delivered 9 workshops on FI to healthcare professionals covering culturally adapted intervention components (gathering information, providing feedback, communication skills, stress management, problem-solving & coping skills and relapse prevention) and guided by a translated manual comprising the accompanying evidence base for each topic and culturally acceptable teaching and learning activities. Researcher training was delivered in a 3 day training course covering symptoms assessment, qualitative process evaluation, practical demonstration, assessment of inter-rater concordance and interviewing skills and techniques. Training in FI and research methods occurred concurrently at the same location but in separate training rooms. The training course was delivered by experienced lecturers, skills trainers and family intervention experts.

### Safety reporting

Adverse events were reported to the Principal Investigator in Indonesia within 24 hours of researchers becoming aware of the event, whether related to the study intervention, for further examination and then to the sponsor and Principal Investigator.

## Process evaluation

We explored intervention implementation, fidelity and experiences of delivering and receiving the intervention in a nested process evaluation using in-depth qualitative interviews post-intervention with a sample of patients, family caregivers and healthcare professionals who took part as trial therapists. Topic guides for carers and patients focused on views of receiving the intervention including content and delivery of each session, perceptions of the usefulness of the intervention, cultural relevance and appropriateness of session content, accessibility of the resources used and views of participating in research including participating in measurement and being randomised. Topic guides for healthcare professionals (trial therapists) focused on exploring factors that influence the delivery of the intervention in practice and views about the usefulness, relevance, and appropriateness of training for trial therapists in delivering the intervention. We explored trial processes in addition to our qualitative process evaluation to evaluate spillover, aiming to identify whether the treatment had unintended effects on individuals outside the intended target group (as described in statistical analysis below) and we evaluated the level of (un)masking by asking assessors to which of the participants they evaluated had participated in the control arm and which had participated in the intervention arm. All interviews were audio recorded, transcribed and translated into English using an independent translator and analysed using thematic analysis [48]. Researchers familiarised themselves with the data, reviewing transcripts to generate initial themes deductively using a priori study aims and inductively to identify themes that fell out with the initial coding framework. Relevant quotations were charted within the deductive themes incorporating inductive themes as the analysis progressed. Analysis was conducted collaboratively by the Indonesian and UK researchers (HS, BAK, RD, S, DW, RF and LR) with the final framework summarised by UK researchers (LR, GA).

## Quantitative analysis

We used descriptive statistics (means, standard deviations, frequencies, percentages) to explore baseline demographic characteristic among service-users and caregivers. We used Cronbach's alpha to evaluate the internal reliability of outcome measures at each stage. In order to assess sensitivity to change of each of the outcome measures used for both service-users and caregivers, we examined within-group change from baseline to assessment at three-month post-completion in both the control and intervention groups. We calculated means, standard deviations, and Cronbach's alpha to assess scale reliability at each time point. We did not test between-group significance due to the nature of the feasibility study. We tested the effectiveness of blinding using a one-sample binomial test, comparing the observed proportion of correct allocation guesses against the proportion we would expect to observe by chance (0.5) to ascertain whether assessors guess differed significantly from random. To assess spillover, we aimed to evaluate between-group differences at both sites but there were unequal group sizes, and insufficient numbers in one site which affects the ability of comparison of means tests to detect true differences between groups so these tests were not completed. Statistical analyses were conducted using SPSS Version 29 (IBM Statistics). Feasibility outcomes are described in Table 1.

## Patient and public involvement

Stakeholder perspectives were utilised extensively in the adaptation of the intervention through consultation groups and consensus work to identify the optimal delivery and format for the intervention. Stakeholders were involved in the research design phase through outreach with our charitable partners. Two members of the Research Advisory Group were service users, one family and one patient member. The trial design was influenced by the Trial Steering Committee in identifying and agreeing optimal trial sites, discussion and collaborating on the design decision to utilise simple randomisation procedures and proposing and agreeing success criteria for transition to a full-scale trial.

**Table 1. Feasibility outcomes reported.**

| Feasibility Outcomes | Measure | Respondent (perspective) | Outcome Reported |
|---|---|---|---|
| Acceptability and satisfaction with the intervention | In-depth qualitative interviews and process evaluation Overall rates of session acceptability by participants and acceptability by therapist, session number/ component and setting | *Family Caregiver Patient Healthcare Professional* | Reported as planned, overall session acceptability reported as overall intervention acceptability due to problems distinguishing intervention components among trial therapists and participants |
| Recruitment, attendance and retention in the intervention | Expected total recruitment and actual total recruitment Recruitment by site (Bogor or Jakarta) and source (self-referral, referred by healthcare professional) Actual numbers approached (number of participants to whom recruitment packs were sent) Numbers not consenting and numbers ineligible Number commencing the intervention [uptake], completing treatment (10 sessions) [retention] obtained via therapist diaries | *Outcome Assessor* | Reported as planned. |
| Completion of outcome measures pre- and post-intervention | Participant age, gender, relationship to service-user, service-user diagnosis, number of hours of contact per week, living status. Report on the distribution of the outcome measures, including assessment of normality. Data completeness includes number of service users/family with complete follow up at each time point, completeness of each separate outcome, at each time point and reasons for incomplete data, if available. Outcome measures (PANSS, FQ, GHQ-28, ECI, KAST, IEQ) In-depth qualitative interviews and process evaluation | *Family Caregiver Patient Healthcare Professional Outcome Assessor* | Reported as planned, with the exception of reasons for incomplete data as there were no missing data. |
| Recruitment of healthcare workers as therapists | Expected total recruitment and actual total recruitment Recruitment by site (Bogor or Jakarta) | *Trial Manager* | Reported as planned, though therapist caseloads are reported for therapists working in pairs due to local decisions to implement buddy system in delivering the intervention |
| Fidelity to the intervention and experience of delivering the intervention | In-depth qualitative interviews and process evaluation FIPAS | *Family Caregiver Patient Healthcare Professional* | FIPAS not reported as this was not collected at the point of intervention delivery. Fidelity was assessed using qualitative interview. |
| Trial processes | Evaluate change on outcomes by group to assess spillover The study is not powered to estimate changes in health outcomes. Masking will be assessed by asking researchers maintaining blinding Satisfaction with trial processes including randomisation, completion of outcome measures, consent procedures, information-giving procedures, data management and storage. | *Family Caregiver Patient Healthcare Professional* | We determined rates of blinding but not spillover. Unmasking evaluated by experience of researchers conducting the interviews. Analysis of covariance could not be conducted to assess spillover due to uneven recruitment between sites. |

## Results

### Participants

The trial planned to recruit 60 dyads but earlier recruitment in one site led to almost complete recruitment, thus recruitment in this site was paused allowing for recruitment in our second site and randomisation was halted at 74 dyads. Inclusion and exclusion criteria for the patients and their family caregivers are listed here (Fig 1). Baseline data measures were distributed to 74 primary carers and 74 patients with schizophrenia. Prior to commencement of

**CONSORT 2025 Flow Diagram**

**Fig 1. CONSORT flow diagram.**

the intervention, four dyads in the intervention group dropped out; one family caregiver died, one was subsequently diagnosed with a long-term physical condition and two were unable to meet the time commitment. Data were analysed for the remaining 70 enrolled in the trial and 33 (n = 47.1%) were randomly assigned to receive our adapted intervention and 37 (52.9%) were assigned to the control group. Baseline characteristics are presented in Table 2. Participants in the qualitative process evaluation comprised 13 stakeholders selected from those who participated in the control or intervention arm, including patients and families and healthcare professionals who were recruited as trial therapists.

**Differences between the sample characteristics of the intervention and control group.** There were significant differences between the ages of the primary carers in the control group and the intervention group $t(70)$ = −1.79, $p$ = .039, with the latter being older. There were no differences between groups in the age of the patients, the total duration of illness and the economic situation of the families in each group. Outcome measures are reported in Table 3.

## Recruitment, attendance and retention in the intervention

We recruited 74 dyads (n = 148) to participate in the feasibility trial; 14 more families than we intended to recruit to ensure that we had representation from both sites. All were successfully randomised to the intervention and the control group

**Table 2. Patient and family characteristics.**

| | | Control (n = 37) | Intervention (n = 33) | Total (n = 70) |
|---|---|---|---|---|
| **Patient** | | Mean (SD) | Mean (SD) | Mean (SD) |
| Age | | 40.7 (10.5) | 38.6 (9.7) | 39.7 (10.1) |
| | | n (%) | n (%) | n (%) |
| Sex | Male | 31 (83.8) | 23 (69.7) | 54 (77.2) |
| | Female | 6 (16.2) | 10 (30.3) | 16 (22.8) |
| Marital status | not married | 31 (83.8) | 27 (81.8) | 58 (82.9) |
| | Married | 3 (8.1) | 4 (12.1) | 7 (10.0) |
| | Widowed | 3 (8.1) | 2 (6.1) | – |
| Highest education | no school | 5 (13.5) | 5 (15.2) | 5 (7.1) |
| | Elementary | 8 (21.6) | 6 (18.2) | 10 (14.3) |
| | junior high | 6 (16.2) | 5 (15.2) | 11 (15.7) |
| | senior high | 18 (48.6) | 15 (45.5) | 33 (47.1) |
| | Diploma | – | 2 (6.1) | – |
| Duration of illness (years) | | 17.5 (9.56) | 14.3 (7.8) | 16.0 (8.9) |
| | | Mean (SD) | Mean (SD) | Mean (SD) |
| Service provider | primary care centre | 29 (78.4) | 21 (63.6) | 50 (71.4) |
| | tertiary services | 8 (21.6) | 12 (36.4) | 20 (28.6) |
| Past hospitalisation | None | 18 (48.6) | 11 (33.1) | 29 (41.4) |
| | >=1 | 19 (51.3) | 22 (66.9) | 41 (68.6) |
| **Family Caregiver** | | Mean (SD) | Mean (SD) | Mean (SD) |
| Age | | 58.9 (9.9) | 63.3 (10.4) | 61.0 (10.3) |
| | | n (%) | n (%) | n (%) |
| Sex | Male | 6 (16.2) | 8 (24.2) | 14 (20.0) |
| | Female | 31 (83.8) | 25 (75.8) | 56 (80.0) |
| Marital Status | not married | – | 8 (24.2) | 49 (70) |
| | Married | 24 (64.9) | 25 (75.8) | – |
| | Widowed | 13 (35.1) | – | 21 (30) |
| Highest Education | no school | 8 (21.6) | 10 (30.3) | 10 (4.3) |
| | Elementary | 13 (35.1) | 14 (42.2) | 14 (20) |
| | junior high | 10 (27.0) | 3 (9.1) | 11 (15.7) |
| | senior high | 6 (16.2) | 6 (18.2) | 33 (47.1) |
| | Diploma | – | – | |
| Relationship to patient | Father | 4 (10.8) | 7 (21.2) | 11 (15.7) |
| | Mother | 20 (54.1) | 16 (48.5) | 36 (51.5) |
| | Sibling | 3 (8.9) | 3 (9.1) | 6 (8.6) |
| | Spouse | 8 (21.6) | 2 (6.0) | 10 (7.0) |
| | extended family | 2 (5.4) | 5 (15.2) | 7 (14.2) |
| Family income | < 2.500.000 IDR | 25 (67.6) | 26 (78.8) | 51 (72.6) |
| | 2.500.000–5.000.000 IDR | 12 (32.4) | 4 (12.1) | 16 (22.9) |
| | >5.000.000 IDR | – | 3 (9.1) | 3 (4.3) |

though four dropped out following allocation due to practical and personal reasons rather than dissatisfaction with the intervention or trial procedures. Retention in the feasibility trial was 95% for the duration of the intervention and this remained the same at post-intervention and 3-month follow-up evaluations. Those who dropped out were allocated to

**Table 3. Outcome measures.**

| Measures | Baseline | | | Post-Intervention | | | Three Month Post-Intervention | | |
|---|---|---|---|---|---|---|---|---|---|
| | Intervention (n = 33) | Control (n = 37) | Scale Reliability | Intervention (n = 33) | Control (n = 37) | Scale Reliability | Intervention (n = 33) | Control (n = 37) | Scale Reliability |
| **Patient** | Mean (SD) | Mean (SD) | Cronbach's α (n) | Mean (SD) | Mean (SD) | Cronbach's α (n) | Mean (SD) | Mean (SD) | Cronbach's α (n) |
| PANSS | 57.0 (20.2) | 66.0 (22.2) | .941 (30) | 54.0 (23.1) | 64.8 (27.4) | .964 (30) | 49.0 (17.1) | 55.7 (21.3) | .958 (30) |
| GAF | 76.2 (12.5) | 76.0 (11.5) | – | 79.1 (12.5) | 77.9 (12.3) | – | 81.9 (13.7) | 77.7 (14.1) | – |
| **Family Caregiver** | | | | | | | | | |
| GHQ | 26.4 (3.7) | 25.6 (4.5) | . 713 (12) | 25.8 (9.2) | 26.4 (6.2) | .937 (12) | 29.3 (3.0) | 27.8 (4.9) | .866 (12) |
| KAST | 6.6 (2.4) | 6.4 (2.5) | .442 (18) | 9.6 (2.4) | 8.5 (2.7) | .548 (18) | 11.1 (2.7) | 9.4 (3.6) | .709 (18) |
| IEQ | 41.4 (17.3) | 46.4 (23.4) | .896 (31) | 29.3 (20.1) | 40.7 (19.9) | .924 (31) | 32.8 (17.5) | 36.4 (19.5) | .898 (31) |
| ECI | 90.6 (27.3) | 100.1 (27.6) | .885 (66) | 67.8 (29.8) | 77.4 (35.5) | .941 (66) | 79.4 (30.7) | 84.6 (38.2) | .941 (66) |
| FQ | 43.1 (8.2) | 44.4 (10.5) | .866 (20) | 36.9 (9.1) | 42.6 (13.6) | .942 (20) | 35.1 (9.5) | 38.2 (14.0) | .950 (20) |

Note: PANSS = Positive and Negative Syndrome Scale, GAF = Global Assessment of Functioning, GHQ = General Health Questionnaire, KAST = Knowledge About Schizophrenia Test, IEQ = Involvement Evaluation Questionnaire, ECI = Experience of Caregiving Inventory, FQ = Family Questionnaire

the intervention group thus retention in the control group was 100% at baseline, post-intervention and 3-month follow-up. Retention in the intervention was 89% at post-intervention and 3-month follow-up. Outcome measures for those who dropped out were removed before analysis was conducted as per ethical procedures.

## Outcome measures pre- and post-intervention

We tested inter-rater reliability between raters prior to undertaking the study on measures of symptomatology. ICC scores, based on a mean-rating (k = 10), absolute-agreement, 2-way mixed-effects model, ranged from 0.87 (CI = 0.77–0.96) to 0.98 (CI = 0.97–0.99) establishing interrater reliability from good to excellent. There was complete outcome measurement at the three measurement points throughout the trial among those who were retained in the trial. Estimates of the internal consistency, alongside descriptive statistics, are reported for each scale at each time point in Table 4. Most scales demonstrate good to excellent consistency at each time point (α >=.8) and one scale demonstrated poor consistency at each time point (KAST, α < .6) [49]. At baseline, over half the sample (n = 38, 54.3%) did not meet remission criteria based on the RSWG for schizophrenia definition of remission [38], scores of mild on the following PANSS items; P1 delusions, P2 conceptual disorganisation, P3 hallucinatory behaviour, N1 blunted affect, N4 social withdrawal, N6 lack of spontaneity, G5 mannerisms and posturing and G9 unusual thought content. A future trial will utilise relapse as a primary outcome measure and this was assessed at three time points throughout the study. Six participants (8.4%) met relapse criteria post-intervention defined as an increase in PANSS total score >=12 points and>=1 point increase together with a score of >4 in>=1 item of the positive and disorganised scales used in the RSWG for schizophrenia definition of remission [38]. No participant met these criteria at three months follow-up.

## Recruitment of healthcare workers as therapists

We recruited 18 therapists to participate in the training and all the recruited healthcare workers participated in and completed the training. There were 13 therapists who subsequently committed to working with a minimum of 1 family for the duration of the study. The therapists comprised mental health nurses with a wide range of training and experience (see Table 4). Therapists worked in pairs and each pair was allocated a minimum of 4 families and a maximum of 8, with the exception of three therapists working together in one site. Five healthcare professionals received training but did not administer the intervention as they were unable to commit to working with the required minimum participants or were

**Table 4. Therapist characteristics and allocations.**

| Therapist | Sex | Occupation/Location | Education | Dyad Allocation | Site |
|---|---|---|---|---|---|
| Therapist 1 | Female | Psychiatric Nurse/Mental Hospital | Bachelor of Nursing, Postgraduate Diploma Mental Health | 8 participant dyads | Jakarta/Bogor |
| Therapist 2 | Male | Community Nurse/Primary Care | Diploma in Nursing | 7 participant dyads | Jakarta/Bogor |
| Therapist 3 | Female | Psychiatric Nurse/Mental Hospital | Bachelor of Nursing, Postgraduate Diploma Mental Health | 8 participant dyads | Jakarta/Bogor |
| Therapist 4 | Female | Psychiatric Nurse/Mental Hospital | Bachelor of Nursing, Postgraduate Diploma Mental Health | 8 participant dyads | Jakarta/Bogor |
| Therapist 5 | Female | Lecturer | Bachelor of Nursing, Postgraduate Diploma Mental Health | 4 participant dyads | Jakarta |
| Therapist 6 | Female | Lecturer | PhD | 4 participant dyads | Bogor |
| Therapist 7 | Female | Community Nurse/Primary Care | Bachelor of Nursing | 4 participant dyads | Bogor |
| Therapist 8 | Female | Lecturer | Bachelor of Nursing, Postgraduate Diploma Mental Health | 5 participant dyads | Bogor |
| Therapist 9 | Female | Community Nurse/Primary Care | Diploma in Nursing | 5 participant dyads | Bogor |
| Therapist 10 | Female | Community Nurse/Primary Care | Bachelor of Nursing, | 4 participant dyads | Bogor |
| Therapist 11 | Female | Psychiatric Nurse/Mental Hospital | Bachelor of Nursing, Postgraduate Diploma Mental Health | 4 participant dyads | Bogor |
| Therapist 12 | Female | Lecturer | Bachelor of Nursing, Postgraduate Diploma Mental Health | 4 participant dyads | Bogor |
| Therapist 13 | Female | Psychiatric Nurse/Mental Hospital | Bachelor of Nursing, Postgraduate Diploma Mental Health | 4 participant dyads | Bogor |

located at a distance from the sites that would make delivering the intervention challenging. A one-sample proportion test comparing the proportion of correct treatment guesses to the chance level of 50% indicated that assessors guessed correctly less often than expected by chance (36.5%, 95% CI 26–48%, p = .020, two-sided). This suggests that blinding was well maintained [50]. Estimation of therapist effect was not possible, as the intervention was delivered by multiple therapists with variable therapist-therapist pairings.

## Process evaluation

Five themes describing the implementation of culturally adapted family interventions and proposed mechanisms of effect were identified using qualitative data obtained from patients and family members who participate in control and intervention arms, healthcare professionals who delivered interventions and in primary care services where patients and families were recruited (see Table 5).

## Acceptance and recognition

The intervention was well-received among participants, garnering positive feedback for its impact on social inclusion and the holistic care approach was valued among recipients. Participants and trial therapists described that the single-family intervention format, delivered in patient's homes was preferred for convenience and comfort. Having an intervention at home gave families a sense of being seen by healthcare professionals, and families expressed gratitude for being visited in their homes, feeling they were being acknowledged and recognised. While some therapists felt this approach an extension of their usual ways of working with service-users, others found the process allowed space and time to really listen and understand families' experiences which then became a profound experience for the therapist. The intervention highlighted a broader shift in how healthcare professionals viewed and approached family involvement having previously been naïve to the needs of families and in this sense, the intervention helped bridge this gap, creating a more family

**Table 5. Thematic analysis of qualitative process data.**

| | Illustrative Quote | Respondent | Participant ID |
|---|---|---|---|
| Acceptance and recognition | 'I'm happy whenever someone comes over to have a talk. I know they are from the health center *[sic]* and want to talk, confide, and so on. Everyone who comes to visit me is kind.' | Patient Control Group | PC1 |
| | 'the family began by sharing their experiences with mental health issues. So the first meeting was more on the sad side, with some family members crying, it was like they were ventilating their stories to the therapist, being asked questions like that. That first meeting was quite impactful in that sense.' | Therapist | T1 |
| | '… and sometimes it is indeed necessary to go directly to the community, and I also asked how they felt, and they were happy. Especially when I asked if there had been any health workers who came like this before, providing education like this, and they said no, they were happy with this arrangement, so we're all happy' | Therapist | T2 |
| | 'when I first learned about this method at the workshop, I couldn't imagine how it would be implemented later, and it didn't really hit home, I didn't feel it in my heart. But then, after going out and applying it to several families, um, I felt that I learned many new things.' | Therapist | T2 |
| | 'it was evident that if applied, it could have a positive impact, especially on families because our focus is on how families are capable of handling, not just managing the patient or the person with the disorder, but also managing themselves, including their emotions, and how to channel their concerns and complaints' | Therapist | T1 |
| | 'It's more about trust from the family or caregivers and the patient giving their trust to the therapist. Like, "Oh yes, this therapist wants to help me." So the first session is crucial. The crucial part is that the therapist is able to convince the family and the patient that "today I am here, we will practice and discuss together that our goal of caring for the person with a mental health disorder and reducing stress is what we will achieve."' | Therapist | T1 |
| | 'We can feel how we have been indifferent to the families, and with this research, it seems we are more focused on families. Previously, we didn't know what kinds of issues the families caring for people with severe mental disorders faced' | Therapist | T4 |
| Increased awareness | 'Now everyone knows the issues, all the problems are out in the open……They know things like I have trouble sleeping, get paranoid, and such. I get extremely paranoid, become quiet, but I need to be left alone.' | Patient Intervention Group | PI1 |
| | 'I liked it because I wanted to know. So, I learned what schizophrenia is, and they explained it. Now I understand better that schizophrenia is like this and that.' | Patient Intervention Group | PI2 |
| | 'It felt good to share, there was progress…. To understand what's inside of my illness' | Patient Intervention Group | PI2 |
| | 'The changes are that he understands more, doesn't rebel….It's very beneficial.' | Family Caregiver Intervention Group | FI1 |
| | 'there are sessions where therapy needs to be provided both to the family and the patient……One hour is not enough for that because you have to address both the family and the patient.' | Therapist | T1 |
| | 'That is in the psychoeducation session….patients in the community should already be proficient since they might have already received training from hospital care or maybe from the community health center, but in practice, or in reality, it's like we have to train them from scratch. So, the time feels insufficient.' | Therapist | T2 |
| Personalisation | 'So, it really goes back to the background or history of the patients and their families. Because their education level…. none of them had an education beyond high school | Therapist | T1 |
| | 'we have to be aware that education levels differ. If one caregiver didn't finish elementary school and another did, we need to adjust accordingly' | Therapist | T3 |
| | 'But in the implementation, of course, we have to modify it, uh, not exactly the same, but we don't deviate from the context……it's not about modifying the intervention itself….It's really the approach that differs. ' | Therapist | T2 |
| Conceptual clarity of individual components | 'so when they were given therapies like, for example, goal setting, they were confused. Or it could also be perhaps because the ability of the therapist is not yet adequate, I also don't know.' | Therapist | T1 |
| | 'it seems to be quite extraordinary. I might even question my ability to fully grasp it, you know, because I indeed have limitations with language or whatever it's called to understand it, right?' | Therapist | T3 |

centred approach. Therapists acknowledged the value of this shift, as it addressed long-standing limitations in healthcare and emphasised the need to educate wider groups of healthcare workers who work directly with families in the community to ensure wider professionals and workers are receptive to family-centred approaches.

### Increasing awareness

Families believed the intervention demonstrated optimism for recovery and a desire for continued support, emphasising the importance of educating and involving families in the therapeutic process while facilitating increased awareness. Improved knowledge of illness, symptoms, ways of self-management and prescribed treatments were reported by service-users as well as finding ways to open up conversations about their experiences with their families bringing new appreciation for their difficulties among their caregivers positively impacting their relationships. Therapists perceived increased knowledge of evidence and theory for family involvement as well as knowledge of family needs. On this latter point, the complexity of information-sharing with both service-users and families was noted as a specific challenge which required careful consideration within the context of the setting, tailoring the intervention to specific family's needs.

### Personalisation

Intergenerational perspectives on mental illness and cultural adaptations in therapy, such as blending traditional and modern treatments, were particularly salient among recipients. All therapists highlighted the need for adaptations based on the structure of families, the number of caregivers and demographics including age and educational background which were key to successful delivery and a degree of personalisation is needed to deliver the intervention at a pace that suited each family. Timing of delivery was key in the context of illness duration and stage, and some therapists queried whether the intervention might be more impactful if delivered at earlier illness stages. Each session was flexibly delivered to suit both family and therapist diaries, though one unintended outcome were changes to the duration of each session. Sessions were designed to last roughly 60 minutes, however, in practice, sessions were at times limited to just 30 minutes which may have impacted the depth of discussions, implementation of cognitive-behavioural approaches and the extent of family engagement with the intervention content, material and resources.

### Conceptual clarity of individual components

Strategies for problem-solving and stress management were highlighted among intervention recipients as useful and valuable elements of the intervention, noted for helping to create a sense of empowerment among families, as they became more confident in their caregiving abilities. They highlighted improvements not only in their ability to manage their relative's condition but also in their own emotional regulation and coping strategies. Conversely, some participants and therapists struggled to differentiate between some components of the intervention itself, making it difficult to implement as originally intended. This issue was particularly evident in conceptual understanding, as some found it hard to distinguish between core elements related to goal-setting and problem-solving, noting poor comprehensibility with the latter in particular.

### Acceptability of trial processes

Overall, the trial processes were well received, with participants expressing appreciation for the structured nature of the assessments, making it easier for both families and therapists to engage effectively with the study. There were roughly equal group sizes between intervention attendees. Families not allocated to the intervention appreciated the support provided by researchers, which made them feel acknowledged and supported. Allocation was concealed to investigators reducing the risk of selection bias and the intervention was delivered as allocated. Concealing the treatment after the allocation has been made is more difficult to achieve. Though they did not ask participants their allocation directly as

instructed, assessors reported the level of knowledge of some participants was qualitatively different, allowing them to infer which participants had been allocated to the intervention, potentially introducing bias.

## Discussion

We aimed to evaluate key feasibility outcomes to assess whether a future full-scale trial of FI for people with schizophrenia is practical and worthwhile rather than testing intervention effectiveness. We included measures of recruitment and retention, intervention acceptability, data collection completeness, resource requirements, and predefined progression criteria for moving to a definitive trial. Recruitment was completed on time and at the intended scale; however, unforeseen delays at one site led to over-recruitment at the partner site and the need to recruit a small number of participants from the second site to enable qualitative exploration of implementation across both settings. We recruited more participants than planned, exceeding our target of 30 dyads per arm. Our conservative sample size was based on an ≤ 80% outcome completion threshold to minimize the risk of progressing with poor retention, which may underestimate feasibility even when the intervention is workable. We offset this by assuming only a 5% probability of proceeding when infeasible and recruiting more participants than planned does mean our study achieved a larger dataset than originally intended, which can strengthen confidence in the precision of feasibility estimates (e.g., outcome completion). Indeed, higher-than-expected recruitment suggests good interest, acceptability, and engagement with the intervention and study processes, supporting feasibility for a future trial.

Factors considered also included the rate of participant recruitment at each site and the recruitment of therapists. In terms of recruitment rate, sites in Bogor achieved the minimum participant numbers at a uniform pace within the study period. However, the Jakarta site experienced delays, resulting in all participants being recruited within a short timeframe, close to randomisation. We met our additional criterion to recruit one therapist per site; however, there are considerably more complex processes in relation to the recruitment and retention of non-specialist healthcare workers requiring more nuanced exploration of factors that support task-sharing approaches and ensuring recruited trial therapists can be trained and retained satisfactorily. Our qualitative process evaluation and peer supervision using therapist diaries indicated that trial therapists generally followed the protocols and delivered the intervention components reliably within the study timeframe. This success may have been supported by structured clinical training and supervision, which have been associated with sustained improvements in healthcare delivery in low-resource settings in evidence from wider literature [51]. Our approach was further supported by training protocols that drew on both experience and evidence, along with expert input during manual development. The training materials were developed collaboratively and participatively, aiming to be culturally relevant and practical. They included clear learning objectives, suitable delivery formats, and real-life case studies to help illustrate the intervention and its potential impact which our qualitative process evaluation indicated were helpful strategies.

Another challenge in conducting randomised controlled trials and ensuring the credibility of findings is unplanned attrition and loss to follow-up, which can lead to incomplete outcome assessments. Even relatively low levels of attrition or missing interval and outcome data are thought to introduce bias and compromise the validity of trial results [52]. Our study experienced minimal attrition, which supports the calculation of appropriate sample sizes and increases confidence that bias can be minimised in future trials. Attrition was substantially lower than that reported in recent meta-analyses of non-pharmacological interventions for people with schizophrenia [53,54]. Two factors may explain this. First, in lower-resource settings like Indonesia, where mental health services are limited and interventions scarce, participants may be more motivated to engage in experimental studies. Second, we implemented assertive engagement strategies, flexible scheduling, home visits, diligent follow-up, and customised the intervention to participants' needs and preferences, likely supporting retention and outcome completion [55,56] meaning this is a plausible explanation [57]. Some differential attrition was observed, which could bias results if dropouts differed systematically from retained participants. However, the small number of dropouts and inability to collect their outcome data precluded assessment of this impact, though the

reasons for drop-out appeared not to be linked to study processes, such as the timing, burden, or complexity of trial procedures, or the intervention itself, since drop-out occurred prior to intervention initiation and outcome assessment. Incidental data suggest drop-out was primarily due to personal reasons though a future trial should consider comprehensive quantitative and qualitative process evaluations to explore how intervention and study factors influence engagement and retention, key considerations for adoption and scaling.

In terms of outcome assessment, although most of the measures we included performed well, one measure demonstrated poor consistency, meaning the scale did not measure the construct of knowledge of schizophrenia in a dependable way. Participants scores may be due to random error rather than true changes in the construct being measured but there was minimal improvement in consistency at each time point, signalling that participants may have found items confusing, irrelevant, or interpreted them inconsistently. This can highlight issues with the measure's clarity, cultural appropriateness, or length. Applying measures of illness knowledge from high-income to low resource settings presents significant challenges due to issues with cultural bias, relevance, literacy levels and educational attainment [58] and can be problematic in the particular construct being measured. We implemented measures to culturally adapt the scales used in our study [47]; however, further refinement may be needed through additional item analysis, qualitative feedback, cognitive interviewing, or more extensive piloting of revised measures which has been demonstrated to achieve greater consistency in measures of knowledge in this setting [59].

Nonetheless, delivering interventions based on cognitive-behavioural approaches can be cognitively demanding, with a high need for supervision and support. Despite evidence of successfully training healthcare workers to implement the intervention and shifting perspectives more generally among practitioners on the importance of active family involvement in treatment, a minority of workers had trouble distinguishing between the varied components of the intervention following training. Therapists views of the intervention indicated a high degree of coherence, in that the relevance and purpose, its proposed benefits and value were clear, and it could be easily distinguished from current practice [60] however, conceptual understanding, and distinguishing between elements such as problem-solving techniques, goal-setting and emotional support strategies was weak in places. We considered that this may be attributable to cultural norms and practices in that South-east Asian cultures often prioritize collective well-being over individual aspirations, influencing goal setting towards goals that benefit the community or family rather than the individual which may explain a degree of uncertainty.

This challenge highlights the need for enhanced theoretical frameworks, incorporating a Theory of Change approach, explicitly mapping out how and why the intervention is expected to achieve its outcomes, for future intervention development. Evidence suggests task-shifting interventions should be accompanied by health system re-structuring to optimise effectiveness, implementation and affordability benefits of improving access to healthcare [61]. This will be particularly salient for any future trial of family interventions for families of people with schizophrenia in Indonesia as there is limited evidence for task-sharing interventions for severe mental illness when compared with common mental disorders [62] and as such, evaluating and identifying patient, intervention and system-level characteristics associated with successful implementation of task-sharing practices for interventions will fill a significant knowledge gap. Future research could then be designed in such a way that allows measurement of the specific contributions of each intervention module, in order to improve targeting and better understand active ingredients. Equally, evaluating implementation domains that influence successful adoption of the intervention is key to identifying core and peripheral elements of the intervention for a more nuanced exploration of the key constituents of the intervention that need to be retained for fidelity and those potentially adaptable or modifiable elements to retain the integrity of the intervention for future testing.

## Conclusion

Our feasibility trial provides encouraging evidence that a full-scale intervention of FI in Indonesia is practical and could be beneficial. Recruitment targets were achieved within schedule, and retention and outcome completion rates were high, exceeding expectations and indicating strong participant engagement and acceptability of the intervention. The

over-recruitment that occurred due to site-specific delays yielded a larger dataset than planned, enhancing the precision of feasibility estimates without compromising study integrity. Therapist recruitment and retention were achieved, and understanding why some could not deliver the intervention will inform planning for future studies. The structured training and supervision model developed for the study appeared to support fidelity and adherence to intervention protocols. Nonetheless, challenges related to the recruitment and retention of non-specialist health workers and the differentiation of intervention components highlight areas requiring further development and contextual adaptation.

Measurement issues, particularly regarding the scale assessing knowledge of schizophrenia, underscore the need for continued refinement and cultural validation of assessment tools in low-resource settings. Despite these challenges, the study demonstrates that task-shifting approaches can be successfully implemented when supported by robust supervision and contextually grounded training. Future research incorporating a clear theoretical framework, such as a Theory of Change, may better articulate mechanisms of action and guide implementation strategies. Additionally, future trials should evaluate the health system factors that influence adoption, fidelity, and sustainability of task-shared family interventions. Overall, this study establishes a solid foundation for a definitive trial and contributes important evidence on the feasibility of implementing and scaling culturally adapted family interventions for people with schizophrenia in low-resource contexts.

## Supporting information

**S1 File. CONSORT extension pilot and feasibility trials checklist.**
(DOC)

**S2 File. Research protocol FUSION V5.**
(DOCX)

## Acknowledgments

The authors would also like to thank the participants, research advisory group and healthcare professionals (listed below) for giving their time and expertise; Yossie Susanti Eka Putri, Ni Made Suci Wahyuni, Junita Lasma, Siti Amrina Rosyada, Yeni Dearni Pinem, Atik Puji Rahayu, Carolina, Neng Esty Winahayu, Ellya Fadlah, Yunita Astriani Hardayani, Ice Yulia Wardani, Henny Kusumawati and Yudi Ahdan Saputra.

## Author contributions

**Conceptualization:** Laoise Renwick, Penny Bee, Karina Lovell, Herni Susanti.

**Data curation:** Laoise Renwick, Dewi Wulandari, Rizqy Fadilah, Raphita Diorarta, Suherman Suherman, Georgia Addison, Herni Susanti.

**Formal analysis:** Laoise Renwick, Helen Brooks, Budi-anna Keliat, Dewi Wulandari, Rizqy Fadilah, Raphita Diorarta, Suherman Suherman, Georgia Addison, Karina Lovell, Herni Susanti.

**Funding acquisition:** Laoise Renwick, Helen Brooks, Budi-anna Keliat, Penny Bee, Karina Lovell, Herni Susanti.

**Investigation:** Laoise Renwick, Budi-anna Keliat, Dewi Wulandari, Rizqy Fadilah, Raphita Diorarta, Penny Bee, Karina Lovell, Herni Susanti.

**Methodology:** Laoise Renwick, Budi-anna Keliat, Dewi Wulandari, Rizqy Fadilah, Raphita Diorarta, Suherman Suherman, Herni Susanti.

**Project administration:** Laoise Renwick, Budi-anna Keliat, Dewi Wulandari, Rizqy Fadilah, Raphita Diorarta, Suherman Suherman, Herni Susanti.

**Resources:** Laoise Renwick, Budi-anna Keliat, Dewi Wulandari, Rizqy Fadilah, Raphita Diorarta, Suherman Suherman, Herni Susanti.

**Supervision:** Laoise Renwick, Helen Brooks, Herni Susanti.

**Writing – original draft:** Laoise Renwick, Georgia Addison, Herni Susanti.

**Writing – review & editing:** Laoise Renwick, Helen Brooks, Budi-anna Keliat, Dewi Wulandari, Rizqy Fadilah, Raphita Diorarta, Suherman Suherman, Georgia Addison, Penny Bee, Karina Lovell, Herni Susanti.

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
