## [Decision Letter · Decision Letter 0]

26 Sep 2025

Dear Dr. Renwick,

Thank you for submitting your manuscript to PLOS ONE. After careful consideration, we feel that it has merit but does not fully meet PLOS ONE’s publication criteria as it currently stands. Therefore, we invite you to submit a revised version of the manuscript that addresses the points raised during the review process.

Reply the academic editor's feedback Reply the reviewer's feedback

We look forward to receiving your revised manuscript.

Kind regards,

Fadwa Alhalaiqa

Academic Editor

PLOS ONE

**Journal Requirements:**

1. When submitting your revision, we need you to address these additional requirements. Please ensure that your manuscript meets PLOS ONE's style requirements, including those for file naming. The PLOS ONE style templates can be found at https://journals.plos.org/plosone/s/file?id=wjVg/PLOSOne_formatting_sample_main_body.pdf and https://journals.plos.org/plosone/s/file?id=ba62/PLOSOne_formatting_sample_title_authors_affiliations.pdf 2. Thank you for stating the following financial disclosure: This paper presents independent research funded by the Medical Research Council (MR/T003987/1) under its Joint Global Health Trials Funding Scheme titled ‘Reducing Relapse for People with Schizophrenia in Jakarta, Indonesia: Developing a culturally‐relevant, evidence‐based Family Intervention’. This represents joint funding from the Department of Health and Social Care (DHSC), the Foreign, Commonwealth & Development Office (FCDO), the Medical Research Council (MRC) and Wellcome Trust.    Please state what role the funders took in the study.  If the funders had no role, please state: "The funders had no role in study design, data collection and analysis, decision to publish, or preparation of the manuscript." If this statement is not correct you must amend it as needed. Please include this amended Role of Funder statement in your cover letter; we will change the online submission form on your behalf. 3. Thank you for stating the following in the Acknowledgments Section of your manuscript: This paper presents independent research funded by the Medical Research Council (MR/T003987/1) under its Joint Global Health Trials Funding Scheme titled ‘Reducing Relapse for People with Schizophrenia in Jakarta, Indonesia: Developing a culturally‐relevant, evidence‐based Family Intervention’. This represents joint funding from the Department of Health and Social Care (DHSC), the Foreign, Commonwealth & Development Office (FCDO), the Medical Research Council (MRC) and Wellcome Trust. Additional support was provided by the Faculty of Nursing at Universitas Indonesia and the University of Manchester. The views expressed are those of the authors and not necessarily those of the Medical Research Council, Universitas Indonesia or the University of Manchester. The authors would also like to thank the participants, research advisory group and healthcare professionals (listed below) for giving their time and expertise; Yossie Susanti Eka Putri, Ni Made Suci Wahyuni, Junita Lasma, Siti Amrina Rosyada, Yeni Dearni Pinem, Atik Puji Rahayu, Carolina, Neng Esty Winahayu, Ellya Fadlah, Yunita Astriani Hardayani, Ice Yulia Wardani, Henny Kusumawati and Yudi Ahdan Saputra. We note that you have provided funding information that is not currently declared in your Funding Statement. However, funding information should not appear in the Acknowledgments section or other areas of your manuscript. We will only publish funding information present in the Funding Statement section of the online submission form. Please remove any funding-related text from the manuscript and let us know how you would like to update your Funding Statement. Currently, your Funding Statement reads as follows: This paper presents independent research funded by the Medical Research Council (MR/T003987/1) under its Joint Global Health Trials Funding Scheme titled ‘Reducing Relapse for People with Schizophrenia in Jakarta, Indonesia: Developing a culturally‐relevant, evidence‐based Family Intervention’. This represents joint funding from the Department of Health and Social Care (DHSC), the Foreign, Commonwealth & Development Office (FCDO), the Medical Research Council (MRC) and Wellcome Trust.   Please include your amended statements within your cover letter; we will change the online submission form on your behalf. 4. Thank you for stating the following in your Competing Interests section:  “The authors declare no conflicts”  Please complete your Competing Interests on the online submission form to state any Competing Interests. If you have no competing interests, please state "The authors have declared that no competing interests exist.", as detailed online in our guide for authors at http://journals.plos.org/plosone/s/submit-now This information should be included in your cover letter; we will change the online submission form on your behalf. 5. Please note that your Data Availability Statement is currently missing the repository name and/or the DOI/accession number of each dataset and a direct link to access each database. If your manuscript is accepted for publication, you will be asked to provide these details on a very short timeline. We therefore suggest that you provide this information now, though we will not hold up the peer review process if you are unable. 6. When completing the data availability statement of the submission form, you indicated that you will make your data available on acceptance. We strongly recommend all authors decide on a data sharing plan before acceptance, as the process can be lengthy and hold up publication timelines. Please note that, though access restrictions are acceptable now, your entire data will need to be made freely accessible if your manuscript is accepted for publication. This policy applies to all data except where public deposition would breach compliance with the protocol approved by your research ethics board. If you are unable to adhere to our open data policy, please kindly revise your statement to explain your reasoning and we will seek the editor's input on an exemption. Please be assured that, once you have provided your new statement, the assessment of your exemption will not hold up the peer review process. 7. Please ensure that you include a title page within your main document. We do appreciate that you have a title page document uploaded as a separate file, however, as per our author guidelines (http://journals.plos.org/plosone/s/submission-guidelines#loc-title-page) we do require this to be part of the manuscript file itself and not uploaded separately. Could you therefore please include the title page into the beginning of your manuscript file itself, listing all authors and affiliations. 8. Please include captions for your Supporting Information files at the end of your manuscript, and update any in-text citations to match accordingly. Please see our Supporting Information guidelines for more information: http://journals.plos.org/plosone/s/supporting-information. 9. If the reviewer comments include a recommendation to cite specific previously published works, please review and evaluate these publications to determine whether they are relevant and should be cited. There is no requirement to cite these works unless the editor has indicated otherwise. 

**Additional Editor Comments:**

Major Issues

1. Data Availability Statement

o Current statement indicates restrictions. PLOS ONE requires full compliance unless legal/ethical exceptions apply. Authors should clarify exactly how data can be accessed, provide repository details (DOI/URL), and avoid vague terms like “available upon request.”

2. Sample Size & Power Justification

o While the feasibility rationale is provided, more explanation is needed on why 30 dyads per arm was chosen, how this aligns with CONSORT extension for pilot/feasibility studies, and whether the final recruitment of 74 dyads exceeded feasibility expectations.

3. Blinding and Bias

o The manuscript acknowledges difficulty maintaining assessor blinding. More detail on how potential bias was minimized is warranted. For example: Did assessors’ guesses of allocation correlate with actual group assignment? How might this have affected outcomes?

4. Measurement Tools

o Some tools (e.g., KAST) demonstrated poor internal consistency (α < .6). Authors should discuss implications for validity of findings, particularly since feasibility relies on measuring outcomes reliably.

5. Ethics & Consent

o Ethics approvals are listed, but the manuscript should specify the form of consent obtained (written, electronic) and confirm alignment with local regulations .

4. Minor Issues

• Clarity in Abstract: Consider shortening and sharpening the abstract by focusing on feasibility outcomes (recruitment, retention, acceptability) and process evaluation highlights.

• Terminology: At times “family interventions” and “psychoeducation” are used interchangeably. Clarify distinction.

• Tables: Some tables are dense. Consider clearer labeling (e.g., specify “Family Caregiver” vs “Patient” outcomes) for easier reader interpretation.

• References: Ensure all references (especially local studies cited for Indonesia) are updated and formatted per PLOS style.

Reviewers' comments:

**Comments to the Author**

1. Is the manuscript technically sound, and do the data support the conclusions?

Reviewer #1: Yes

2. Has the statistical analysis been performed appropriately and rigorously?

Reviewer #1: Yes

3. Have the authors made all data underlying the findings in their manuscript fully available?

Reviewer #1: No

4. Is the manuscript presented in an intelligible fashion and written in standard English?

Reviewer #1: Yes

**Reviewer #1: ** The paper reports on a feasibility or pilot randomized controlled trial (RCT) assessing the implementation of Family Intervention (FI) for the treatment of schizophrenia. The authors highlight the need for cultural adaptation of FI, particularly in low- and middle-income countries (LMICs) such as Indonesia. This trial aimed to investigate the feasibility of delivering a culturally adapted version of FI to families of individuals with schizophrenia in Indonesia, with the goal of understanding the tasks, resources, and processes involved in scaling up FI in future trials to enable real-world impact.

The manuscript is well-written, with a clear focus on key feasibility outcomes including acceptability, recruitment, retention, and fidelity to intervention components and delivery. The paper correctly does not focus on statistical significance. I have the following minor suggestions for improvement:

Randomization: Please describe the randomization procedure more clearly, ideally in a separate paragraph or section. This should include details on allocation concealment and any stratification used.

Therapist Effects: It appears that the intervention is delivered by therapists, and that individual therapists may be involved with multiple family dyads. Please clarify whether therapist effects are a consideration in your analysis, and if so, how they are being assessed or controlled for.

Attrition: The intervention group had four dropouts, while the control group had none. Could you elaborate on whether these dropouts were related to any specific components of the intervention? A brief explanation would enhance understanding of the intervention’s acceptability and potential burden.

Primary Outcome: Please clearly specify which outcome is considered primary, and whether it is measured cross-sectionally (at a single time point) or longitudinally (over time). This will help readers interpret the focus and structure of the feasibility assessment.

**Do you want your identity to be public for this peer review?** For information about this choice, including consent withdrawal, please see our Privacy Policy

Reviewer #1: No

---

## [Author Response · Author response to Decision Letter 1]

11 Nov 2025

Response to Academic Editors Feedback

We would like to sincerely thank the editor for their careful reading of our manuscript and for the thoughtful, constructive feedback provided. We greatly appreciate the time and expertise invested in offering detailed comments that we believe have helped us to strengthen the quality, clarity, and contribution of our paper.

In revising the manuscript, we have carefully considered all suggestions and have made changes to address the editors concerns relating to the journal requirements and methodological queries regarding the conduct of our feasibility study.

Editor comments are presented in italics below, followed by our responses and descriptions of the corresponding changes made in the revised manuscript. We do hope that the revisions satisfactorily address all concerns and that the improved manuscript meets the expectations of the editor.

Journal Requirements

https://journals.plos.org/plosone/s/file?id=wjVg/PLOSOne_formatting_sample_main_body.pdf [journals.plos.org] and

https://journals.plos.org/plosone/s/file?id=ba62/PLOSOne_formatting_sample_title_authors_affiliations.pdf [journals.plos.org]

We have revised the manuscript to ensure that it meets PLOS ONE’s style requirements including the presentation of headings and captions. We have added the acknowledgements to the manuscript. The captions have been listed as supplementary material below the references.

This paper presents independent research funded by the Medical Research Council (MR/T003987/1) under its Joint Global Health Trials Funding Scheme titled ‘Reducing Relapse for People with Schizophrenia in Jakarta, Indonesia: Developing a culturally‐relevant, evidence‐based Family Intervention’. This represents joint funding from the Department of Health and Social Care (DHSC), the Foreign, Commonwealth & Development Office (FCDO), the Medical Research Council (MRC) and Wellcome Trust.

We have added the following sentence in the financial disclosure for the funding statement. The funders had no role in study design, data collection and analysis, decision to publish, or preparation of the manuscript.

This paper presents independent research funded by the Medical Research Council (MR/T003987/1) under its Joint Global Health Trials Funding Scheme titled ‘Reducing Relapse for People with Schizophrenia in Jakarta, Indonesia: Developing a culturally‐relevant, evidence‐based Family Intervention’. This represents joint funding from the Department of Health and Social Care (DHSC), the Foreign, Commonwealth & Development Office (FCDO), the Medical Research Council (MRC) and Wellcome Trust. Additional support was provided by the Faculty of Nursing at Universitas Indonesia and the University of Manchester. The views expressed are those of the authors and not necessarily those of the Medical Research Council, Universitas Indonesia or the University of Manchester. The authors would also like to thank the participants, research advisory group and healthcare professionals (listed below) for giving their time and expertise; Yossie Susanti Eka Putri, Ni Made Suci Wahyuni, Junita Lasma, Siti Amrina Rosyada, Yeni Dearni Pinem, Atik Puji Rahayu, Carolina, Neng Esty Winahayu, Ellya Fadlah, Yunita Astriani Hardayani, Ice Yulia Wardani, Henny Kusumawati and Yudi Ahdan Saputra.

This paper presents independent research funded by the Medical Research Council (MR/T003987/1) under its Joint Global Health Trials Funding Scheme titled ‘Reducing Relapse for People with Schizophrenia in Jakarta, Indonesia: Developing a culturally‐relevant, evidence‐based Family Intervention’. This represents joint funding from the Department of Health and Social Care (DHSC), the Foreign, Commonwealth & Development Office (FCDO), the Medical Research Council (MRC) and Wellcome Trust.

Please add the following text to the funding statement.

This paper presents independent research funded by the Medical Research Council (MR/T003987/1) under its Joint Global Health Trials Funding Scheme titled ‘Reducing Relapse for People with Schizophrenia in Jakarta, Indonesia: Developing a culturally relevant, evidence‐based Family Intervention’. This represents joint funding from the Department of Health and Social Care (DHSC), the Foreign, Commonwealth & Development Office (FCDO), the Medical Research Council (MRC) and Wellcome Trust. Additional support for this study was provided by the Faculty of Nursing at Universitas Indonesia and the University of Manchester. The views expressed are those of the authors and not necessarily those of the Medical Research Council, Universitas Indonesia or the University of Manchester. The funders had no role in study design, data collection and analysis, decision to publish, or preparation of the manuscript.

“The authors declare no conflicts”

Please complete your Competing Interests on the online submission form to state any Competing Interests. If you have no competing interests, please state "The authors have declared that no competing interests exist.", as detailed online in our guide for authors at http://journals.plos.org/plosone/s/submit-now [journals.plos.org]

Could you please amend the online submission form to state the following: The authors have declared that no competing interests exist.

5. Please note that your Data Availability Statement is currently missing the repository name and/or the DOI/accession number of each dataset and a direct link to access each database. If your manuscript is accepted for publication, you will be asked to provide these details on a very short timeline. We therefore suggest that you provide this information now, though we will not hold up the peer review process if you are unable.

The anonymised dataset is available for restricted data sharing and access will be restricted to authenticated researchers who provide verifiable institutional affiliations and have ethical approval in place for their projects. Data will be available here https://figshare.com/s/47fd98a0334d852ef24b. Please update the online system with this information.

As above data are made available in an anonymised format. In line with the data management plan that was approved during ethical review, data will not be made available to other researchers until primary outputs are published and impact has been reached which will be within three years of generating the dataset. Thereafter, the anonymised dataset is available for restricted data sharing and access will be restricted to authenticated researchers who provide verifiable institutional affiliations and have ethical approval in place for their projects. Data will be available here https://figshare.com/s/47fd98a0334d852ef24b. Please update the online system with this information.

7. Please ensure that you include a title page within your main document. We do appreciate that you have a title page document uploaded as a separate file, however, as per our author guidelines (http://journals.plos.org/plosone/s/submission-guidelines#loc-title-page [journals.plos.org]) we do require this to be part of the manuscript file itself and not uploaded separately.

The title page has been added to the manuscript.

8. Please include captions for your Supporting Information files at the end of your manuscript, and update any in-text citations to match accordingly. Please see our Supporting Information guidelines for more information: http://journals.plos.org/plosone/s/supporting-information [journals.plos.org].

Captions for supporting information files have been added at the end of the manuscript.

No specific published works have been recommended for inclusion in reviewer comments.

Additional Editor Comments:

Major Issues

1. Data Availability Statement

o Current statement indicates restrictions. PLOS ONE requires full compliance unless legal/ethical exceptions apply. Authors should clarify exactly how data can be accessed, provide repository details (DOI/URL), and avoid vague terms like “available upon request.”

We have updated the data availability statement and requested that this be updated in the online submission system. In line with the data management plan that was approved during ethical review, data will not be made available to other researchers until primary outputs are published, and impact has been reached which will be within three years of generating the dataset. Thereafter, the anonymised dataset is available for restricted data sharing and access will be restricted to authenticated researchers who provide verifiable institutional affiliations and have ethical approval in place for their projects. Data will be available here https://figshare.com/s/47fd98a0334d852ef24b. Please update the online system with this information.

2. Sample Size & Power Justification

o While the feasibility rationale is provided, more explanation is needed on why 30 dyads per arm was chosen, how this aligns with CONSORT extension for pilot/feasibility studies, and whether the final recruitment of 74 dyads exceeded feasibility expectations.

We have now provided the feasibility rationale and explained our sample size estimates aligned with the CONSORT extension advising that the number of participants should be provided based on feasibility estimates alongside the rationale. We have also discussed whether final recruitment numbers exceeded expectations in the discussion.

3. Blinding and Bias

o The manuscript acknowledges difficulty maintaining assessor blinding. More detail on how potential bias was minimized is warranted. For example: Did assessors’ guesses of allocation correlate with actual group assignment? How might this have affected outcomes?

We have now provided more detail on the risk of detection bias due to assessors estimation of allocation to intervention and control arms in the methods and in the results where we report the results of a one-sample proportion test comparing the proportion of correct treatment guesses to the chance level of 50%. We advise that estimation of therapist effect was not possible, as the intervention was delivered by multiple therapists with variable therapist-therapist pairings and we discuss the implications of both of these factors in the discussion section.

4. Measurement Tools

o Some tools (e.g., KAST) demonstrated poor internal consistency (α < .6). Authors should discuss implications for validity of findings, particularly since feasibility relies on measuring outcomes reliably.

We have discussed poor internal consistency of some of the tools (i.e. KAST) and the implications of this for the utility of these scales for future studies in the discussion section.

5. Ethics & Consent

o Ethics approvals are listed, but the manuscript should specify the form of consent obtained (written, electronic) and confirm alignment with local regulations.

Consent information is provided on page 8, lines 5 and 6. I have added that the procedures aligned with local ethical and information governance regulations.

4. Minor Issues

• Clarity in Abstract: Consider shortening and sharpening the abstract by focusing on feasibility outcomes (recruitment, retention, acceptability) and process evaluation highlights.

I have revised the abstract for conciseness and brevity and emphasise feasibility outcomes in the methods and reporting of results.

• Terminology: At times “family interventions” and “psychoeducation” are used interchangeably. Clarify distinction.

I have revised the manuscript for clarity. Psychoeducation is used at times as it was a specific component of FI which is reported in our intervention development paper. (Renwick L, Susanti H, Keliat B-a, Wulandari D, Suherman, Fadilah R, et al. Culturally adapting family interventions for people with schizophrenia in Indonesia: An intervention development study using programme theory. International Journal of Nursing Studies Advances. 2025;9:100409.)

• Tables: Some tables are dense. Consider clearer labeling (e.g., specify “Family Caregiver” vs “Patient” outcomes) for easier reader interpretation.

To aid interpretation, I have added a column to table 1 to indicate who collected or reported the data and from which perspective. In table 2 and table 3, patient and family caregiver data are reported separately. In table 5, I have removed the legend and added a column indicating the respondent from whom each quote came.

• References: Ensure all references (especially local studies cited for Indonesia) are updated and formatted per PLOS style.

I have altered the referencing style to Vancouver and updated and formatted for the PLOS style.

Response to Reviewer

We would like to sincerely thank the reviewer for their careful reading of our manuscript and for the thoughtful, constructive feedback provided. We greatly appreciate the time and expertise invested in offering detailed comments that we believe have helped us to strengthen the quality, clarity, and contribution of our paper.

In revising the manuscript, we have carefully considered all suggestions and have made changes to address these concerns. Each concern raised is addressed below with corresponding pages identified whe

---

## [Editor Report · Decision Letter 1]

23 Nov 2025

Delivering culturally adapted family interventions for people with schizophrenia in Indonesia: A feasibility randomised controlled trial and nested process evaluation

PONE-D-25-30156R1

Dear Dr. Laoise Renwick,

We’re pleased to inform you that your manuscript has been judged scientifically suitable for publication and will be formally accepted for publication once it meets all outstanding technical requirements.

Kind regards,

Fadwa Alhalaiqa

Academic Editor

PLOS ONE
---

## [Editor Report · Acceptance letter]

PONE-D-25-30156R1

PLOS One

Dear Dr. Renwick,

I'm pleased to inform you that your manuscript has been deemed suitable for publication in PLOS One. Congratulations! Your manuscript is now being handed over to our production team.

Kind regards,

on behalf of

Pro Fadwa Alhalaiqa

Academic Editor

PLOS One